# Antioxidative Function of Zinc and Its Protection Against the Onset and Progression of Kidney Disease Due to Cadmium

**DOI:** 10.3390/biom15020183

**Published:** 2025-01-27

**Authors:** Soisungwan Satarug

**Affiliations:** Centre for Kidney Disease Research, Translational Research Institute, Woolloongabba, Brisbane, QLD 4102, Australia; sj.satarug@yahoo.com.au

**Keywords:** antioxidant defense, bilirubin, cadmium, heme oxygenase-1, hypertension, kidney disease, zinc deficiency

## Abstract

Chronic kidney disease (CKD) is now the world’s top seventh cause of death from a non-communicable disease, and its incidence is projected to increase further as its major risk factors, including obesity, diabetes, hypertension, and non-alcoholic fatty liver disease (NAFLD), continue to rise. Current evidence has linked the increased prevalence of CKD, diabetes, hypertension, and NAFLD to chronic exposure to the metal pollutant cadmium (Cd). Exposure to Cd is widespread because diet is the main exposure route for most people. Notably, however, the health risk of dietary Cd exposure is underappreciated, and the existing tolerable exposure guidelines for Cd do not afford health protection. New health-protective exposure guidelines are needed. From one’s diet, Cd is absorbed by the intestinal epithelium from where it passes through the liver and accumulates within the kidney tubular epithelial cells. Here, it is bound to metallothionine (MT), and as it is gradually released, it induces tubular damage, tubulointerstitial inflammation and fibrosis, and nephron destruction. The present review provides an update on our knowledge of the exposure levels of Cd that are found to be associated with CKD, NAFLD, and mortality from cardiovascular disease. It discusses the co-existence of hypertension and CKD in people environmentally exposed to Cd. It highlights nuclear and mitochondrial targeting and zinc deficiency as the universal cytotoxic mechanisms of Cd. Special emphasis is placed on the novel antioxidative function of zinc involving de novo heme biosynthesis and the induced expression of heme oxygenase-1 (HO-1). Other exogenous biomolecules with promising anti-Cd toxicity are highlighted.

## 1. Introduction

Environmental exposure to the metal pollutant cadmium (Cd) is one of the most significant public health threats because of its ubiquitous presence in virtually all food types [1,2,3]. Polluted air and cigarette smoke are other two common environmental sources of Cd [4,5,6]. Cd has no nutritional value or physiological role, and it is highly toxic [7]. Because there is no physiologic excretory mechanism, most assimilated Cd is retained within cells, tissues, and organs throughout the body, notably, the kidney, which is the principal site of Cd accumulation and toxicity [7]. In the absence of an elimination mechanism, some of the Cd accumulated in the kidneys increases with age, whilst the kidney Cd burden is essentially determined by the intestinal absorption rate of Cd [7].

According to the WHO’s global health report (https://www.who.int/news-room/fact-sheets/detail/the-top-10-causes-of-death) (accessed on 1 December 2024), the world’s top cause of death in 2019 was ischemic heart disease, while mortality from kidney disease rose from the 13th in 2000 to the 10th in 2019. A dose–response meta-analysis of data from 26 studies observed that the risks of heart failure, coronary heart disease, and stroke were all increased with Cd exposure levels [8]. A blood Cd level of 1 μg/L and a urinary Cd excretion rate of 0.5 μg/g of creatinine may increase the risk of having cardiovascular disease (CVD) by 2.58-fold and 2.79-fold, respectively [8]. These blood and urinary levels were found in a significant proportion of people exposed to Cd in a normal diet, exemplified below.

In a study on the general U.S. population, urinary Cd excretion rates of >1, >0.7, and >0.5 μg/g of creatinine were found in 2.5%, 7.1%, and 16% of non-smoking women aged 20 years or older [9]. In a study from Thailand, 22.5% of non-smoking women, who had low body iron stores, had a Cd excretion rate of 2 µg/g of creatinine [10]. These U.S. and Thai population data indicate that the proportions of people at risk of adverse health effects of Cd are not negligible. Also, they underscore the significance of the body content of metals, notably, iron and zinc, to protect against Cd absorption at high rates.

Evidence that links an enhanced risk of having chronic kidney disease (CKD) with environmental Cd exposure is compelling [11,12,13,14]. Concerningly, the mortality among those with CKD rose with an elevated exposure to environmental Cd, reflected by Cd excretion rates of ≥0.60 μg/g of creatinine or blood Cd levels of ≥0.70 μg/L [15]. Another public health concern is that the Cd excretion rate of 0.60 μg/g of creatinine is 8.7-fold below a toxicity threshold level of 5.24 μg/g of creatinine. This toxicity threshold level of Cd was derived from a risk assessment model using tubular proteinuria as an endpoint [16,17].

The present review is focused on the mechanisms of CKD onset and its progression in people environmentally exposed to Cd. Differences among people in their capacity to absorb and accumulate Cd are highlighted. A toxicological risk assessment in current practice is discussed to argue for a need to develop a new health-protective exposure limit for Cd. The universal cytotoxic mechanisms of Cd targeting mitochondria and zinc deficiency are discussed together with the fundamental anti-Cd defenses, involving zinc, de novo heme biosynthesis, and heme oxygenase-1 (HO-1) induction. Preclinical studies on the potential use of phytochemicals to mitigate the cytotoxicity of Cd are highlighted.

## 2. Transport of Cadmium from the Gut to the Kidneys

### 2.1. Multiple Metal Transport Proteins Involved in the Assimilation of Cadmium from Foods

For most people, exposure to Cd is unavoidable because of its ubiquitous presence in the human diet and polluted air (Figure 1).

Evidence that Cd is present in virtually all food types can be found in Table 1, where the maximal permissible content (MPC) of Cd in foodstuffs has been Cd determined [18].

Because the body cannot synthesize nor degrade any metals, diet is an essential exogenous source of all metal nutrients the body requires such as calcium, zinc, manganese, iron, copper, and cobalt (Ca, Zn, Mn, Fe, Cu, and Co) [19,20,21]. These metals are assimilated from the gut through specialized transport proteins [22,23,24,25]. Unfortunately, however, these metal transporters and pathways also provide an entry route for Cd. Furthermore, Cd in food which forms complexes with metal binding proteins, metallothionine (MT) and phytochelatin (PC), denoted as CdMT and CdPC, can be absorbed through transcytosis [26] and endocytosis [27,28].

Table 2 enlists metal transport proteins responsible for Cd assimilation.

### 2.2. Cadmium Absorption, Renal Accumulation, and Urinary Excretion

The rate of intestinal absorption of an individual metal nutrient is generally low (less than 5%) because each metal nutrient can be taken up by one or two highly specific transport proteins. This selectivity has evolved to prevent toxicity from overloading. In comparison, several transport proteins have been shown to mediate Cd assimilation, and consequently, Cd is absorbed at a much higher rate than each individual metal nutrient. The reported absorption rates of Cd among Japanese women ranged between 24 and 45% [51,52]. Concerningly, however, a conventional toxicological risk assessment assumed Cd absorption rates of 3–7%, resulting in miscalculations and underestimations of the impact of Cd on the function of kidneys [16].

Only a miniscule amount of Cd (0.001–0.005% of the total body content) is excreted each day in urine [53,54]. An extremely slow excretion rate means that most acquired Cd is retained in the body, and hence, the intestinal absorption rate essentially determines the amount of Cd accumulated in the body (termed body burden).

The absorption rate of Cd will increase when the body content of metal nutrients is low and when one’s diet is deficient in these metal nutrients. Indeed, zinc and body iron status show inverse relationships with Cd body burden [55]. Lower body iron stores are associated with higher urinary and blood Cd levels in children [56], adolescent females [57], and women of reproductive age [58,59]. Universally, women have lower body iron stores and higher blood and urinary Cd levels compared to men of a similar age [60,61,62,63]. Habitual consumption of high-Cd-containing foods is another important determinant of the body burden of Cd [64,65,66,67].

A long half-life of Cd is another notable consequence of its extremely slow excretion rate. In Swedish workers exposed to a relatively high dose of Cd, the estimated half-life of Cd ranged from 7.4 to 16 years [68]. A half-life of Cd in non-smoking Swedish subjects exposed to a low dose of Cd in a normal diet was estimated to be 30 years [53,54]. Studies from Japan estimated the half-life of Cd to be 23.4 and 12.4 years in those with urinary Cd concentrations of <5 and >5 µg/L, respectively [69,70]. These data indicate that the lower the body burden, the longer the half-life of Cd.

An estimated half-life of Cd in kidneys was 45 years in another analysis that used data on Cd levels in whole blood, plasma, kidney cortex, and urine samples from Swedish kidney transplant donors (n = 82) [71]. This kidney Cd half-life figure was obtained from a physiologically based toxicokinetic model that incorporated the respective daily systemic Cd uptake at 0.0063 and 0.0085 μg/kg b.w. in men and women and a daily uptake of 1.2 μg for each pack year of smoking [71].

### 2.3. The Excretion of Cadmium Is a Manifestation of Its Cytotoxicity at the Present Time

It is important to note that most excreted Cd emanates from injured or dying tubular cells [72], which means that the excretion of Cd signifies tubular toxicity of Cd at the present time, not the risk of injury in the future [7,72].

For the same reason, the presence of the N-acetyl-β-D-glucosaminidase (NAG) enzyme in urine is indicative of kidney tubular injury as it is released into tubular lumen following the damage or death of tubular cells [73,74]. A dose–response relationship between urinary NAG and urinary Cd was reported in at least 30 publications [75].

A substantial tubular damage appeared to occur at a low Cd body burden, which produced a urinary Cd as low as 0.3 μg/g of creatinine. In a study from the United Kingdom (U.K.), the probability of having an elevated urinary NAG excretion rose 2.6-fold and 3.6-fold at urinary Cd excretion rates of 0.3 and 0.5 μg/g of creatinine, respectively [76]. In a Thai population cohort study [77], a net loss of tubular cells per nephron was apparent as Cd exposure continued and tubular proteinuria ensued, indicated by the β_2_-microglobulin (β_2_M) excretion exceeding 300 µg/g of creatinine. This tubular proteinuria is a manifestation of severe pathologies, resulting in rapid kidney functional deterioration [78,79,80,81]. Thus, it appears illogical to define a dietary exposure limit and a toxicity threshold level of Cd based on the β_2_M excretion of 300 µg/g of creatinine as an endpoint. Further discussion can be found in Section 4.1.

## 3. Mortality Risk and Liver and Kidney Diseases in Low-Dose-Exposure Scenarios

This section highlights data from U.S. population studies, known as the National Health and Nutrition Examination Survey (NHANES), which provides data on exposure levels of more than 200 environmental chemicals, Cd included, experienced by the general U.S. population and their potential adverse health effects [82]. Here, the Cd exposure levels in the U.S. associated with mortality and risks of kidney and liver diseases are listed in Table 3.

Low environmental Cd exposure in the U.S. contributes to mortality from CVD and all causes [83,84,85]. It also contributes to the increased prevalence of CKD and NAFLD [86,87]. These adverse health effects of Cd on the liver and kidneys were observed at a very low body burden, and they are both highly prevalent globally.

### 3.1. Diagnosis and Staging of Chronic Kidney Disease

CKD is a progressive disease, which is diagnosed when the eGFR falls below 60 mL/min/1.73 m^2^ (low eGFR) or when there is albuminuria which persists for at least 3 months [88,89,90]. CKD is now the 7th top cause of global mortality, and it will be the 5th leading cause of years of life lost by 2040 [91,92]. The incidence of kidney disease is projected to increase further as its major risk factors, including obesity, type 2 diabetes, hypertension, and non-alcoholic fatty liver disease, continue to rise [88,89,90,91,92].

CKD stages 1, 2, 3, 4, and 5 correspond to eGFRs of 90–119, 60–89, 30–59, 15–29, and <15 mL/min/1.73 m^2^, respectively [88,89,90]. CKD reaches an end stage when eGFR falls below 15 mL/min/1.73 m^2^, at which time dialysis or a kidney transplant is required for survival [90]. In its early stages, CKD is largely asymptomatic. This makes its early detection difficult and the initiation of early treatment, which can significantly prevent CKD progression, limited.

### 3.2. CKD and Hypertension in People Chronically Exposed to Cadmium

Enhanced risks of kidney damage [93,94,95], albuminuria [96,97,98], proteinuria [98,99], and a low eGFR [11,12,13,14] have repeatedly been linked to chronic environmental Cd exposure. Proteinuria predicts the continued progressive decline of eGFR [100,101,102,103,104].

The Fukuoka Kidney Disease Registry Study, from Japan, reported that advanced kidney fibrosis was found in a higher frequency among those with a higher liver Fibrosis-4 index [105]. Similarly, a Scottish prospective cohort study of 2046 persons, aged ≥18 years, reported a 1.31-fold increase in the risk of developing CKD among those with liver fibrosis who showed no evidence of structural, autoimmune, or malignant CKD [106]. The development of hypertension associated with liver fibrosis may explain a connection between Cd-induced liver fibrosis and an increased CKD risk [87].

Hypertension, defined as a systolic blood pressure (SBP) of ≥140 mmHg or a diastolic blood pressure (DBP) of ≥90 mmHg, is another highly prevalent ill-health condition worldwide [107]. Blood Cd levels of ≥0.80 μg/L were associated with an increased mortality from CVD in non-smokers who had hypertension [108]. Cd-linked hypertension has consistently been noted in many general populations, including the U.S. [109,110,111], Canada [112], China [113,114,115], Korea [116,117], and Japan [118]. In a systematic review and a dose–response meta-analysis, the risk of hypertension rose with blood and urinary Cd levels [119].

### 3.3. An Inverse Relationship of Blood Pressure and eGFR

The connection between kidney damage and hypertension development can be inferred from the indispensable role of kidneys in long-term blood pressure regulation [120,121]. Satarug et al. (2024) investigated the relationship between blood pressure and eGFR in a Thai population cohort study [122]. They reported that there was a two-fold increase in the prevalence of hypertension at a urinary Cd excretion rate of 0.98 µg/g of creatinine, or a blood Cd level of 0.61 µg/L, while SBP showed an inverse relationship with eGFR in both women (β = −0.227) and men (β = −0.320) who had an elevated body burden of Cd (Figure 2a). DBP showed a weak correlation with eGFR (Figure 2b).

Through a mediation analysis, a rising SBP and DBP were found to be due to eGFR reduction induced by Cd (Figure 3). Therefore, eGFR appeared to be a full mediator of rising SBPs and DBPs among subjects with an elevated body burden of Cd [122].

Ischemic acute tubular necrosis and acute and chronic tubulointerstitial fibrosis create impediments to filtration, and thus, they cause a reduction in eGFR [123,124,125]. Consequently, the kidneys eliminate less water and sodium, which then induces blood pressure increments. Increased sodium retention and reduced sodium excretion were observed in rats with Cd-induced hypertension [126,127,128]. Thus, Cd exposure may increase blood pressure and hypertension risk via an enhanced tubular avidity for filtered sodium.

In summary, the data presented in this section indicate that low environmental Cd exposure in the U.S. increased mortality from all causes and from CVD [15,83,84,85]. It also increased the prevalence of both liver and kidney diseases [12,87,88]. Incident CKD and NAFLD may reflect the continuation of toxic environments.

The benchmark dose analysis data indicate that a conventional toxicological risk assessment of dietary Cd exposure incorporated imprecisions which bias the dose–response relationship toward the null [7,122,129]. New health-protective exposure guidelines should be developed. To this end, a fall of eGFR by 5–10% can be used as a sensitive endpoint for estimating a safe dietary Cd exposure level, instead of β_2_M excretion exceeding 300 µg/g of creatinine. An exposure limit for dietary Cd, derived from eGFR reduction as an endpoint, will preserve the functional integrity of the liver and kidneys while minimizing the risk of hypertension that is associated with eGFR loss.

## 4. The Nephrotoxicity of Cadmium and Protective Effects of Zinc

Approximately 8–13% of the world’s population is living with CKD [91,92]. In its early stages, CKD is asymptomatic, and it is diagnosed when there is a substantial loss of functioning nephrons, evident from a fall of eGFR to one third of the normal range. This diagnostic stage often co-exists with disease comorbidities such as hypertension and proteinuria [88,89,90,130]. As is typical, reductions in eGFR are irreversible and are likely to decline further to 15 mL/min/1.73 m^2^, which marks end-stage kidney disease, a condition that requires dialysis or a kidney transplant for survival.

This section focuses on the contribution of Cd exposure to both the onset and progression of CKD and protective effects of the metal nutrient zinc and antioxidants of plant origin.

### 4.1. Manifestation of the Nephrotoxicity of Cadmium

Kidney disease associated with chronic environmental Cd exposure is primarily due to proximal tubular cell damage and malfunction. This results in a sustained decline in eGFR, hypertension, and proteinuria. A proposed pathogenesis of Cd-induced CKD and progression to end-stage kidney disease is presented in Table 4.

Tubular cell injury, indicated by an increased excretion of NAG, β_2_M, RBP, and KIM-1, are the most frequently reported effects of chronic environmental Cd exposure [7,93,94,95]. A study from Taiwan suggested an increased KIM-1 excretion to be a more sensitive indicator of Cd toxicity in CKD patients than conventional markers [131].

With a continuing Cd influx, tubular cell damage and cell death are intensified, and nephrons are destroyed. The tubule injury reduces the reabsorption of proteins, leading to an appearance of protein in the urine. A dose–response analysis informed a 5% increase in total protein excretion at a urinary Cd excretion of 0.0536 µg/g of creatinine [122]. The loss of functioning nephrons causes eGFR to fall further. SBP and DBP rise as eGFR falls (Figure 3), and hypertension develops. The risk of having hypertension doubled at a urinary Cd excretion of 0.98 µg/g of creatinine [122].

Hypertension is one of the most widely recognized consequences of kidney damage [120,132,133]. An increased risk of hypertension was linked with tubular malfunction, reflected by urinary β_2_M excretion rates of ≥145 μg/g of creatinine, compared with β_2_M excretion rates of ≤84.5 μg/g of creatinine [78]. At severe tubular malfunction (β_2_M excretion rates of ≥300 µg/g of creatinine), there was a 79% increase in the risk of eGFR deterioration at high rates, i.e., 10 mL/min/1.73 m^2^ in 5 years [79].

In summary, kidney disease associated with chronic environmental Cd exposure is primarily due to proximal tubule damage and malfunction, leading to nephron destruction, a decrease in eGFR, and hypertension [77,80,121,122].

### 4.2. The Kidney Tubule as the Principal Target of Cadmium Toxicity

Figure 4 depicts the kidney tubular epithelial cell, the principal Cd accumulation site and toxicity.

As Figure 4 depicts, kidney tubular epithelial cells are well equipped with many metal transport proteins and receptor-mediated systems for the internalization of whole proteins. These are for the retrieval of all nutrients, including the proteins albumin and transferrin, and the essential metals Zn and Fe [48,49,50]. Most of these transport pathways and systems also facilitate Cd entrance. ZIP8, ZIP10, and ZIP14 mediate Cd uptake [18,134,135].

Transgenic mice with three more copies of the ZIP8 gene accumulated twice as much Cd in the kidneys following oral Cd exposure, and the proximal tubular cells from these mice had elevated levels of ZIP8 on the apical membrane, which explained their high sensitivity to Cd toxicity [18]. The expression of the ZIP8 protein by human proximal tubule cells has been shown [136].

### 4.3. The Cytotoxic Mechanisms of Cadmium

The proximal tubular epithelial cells of the kidneys are particularly rich in mitochondria, and their homeostasis and survival depend heavily on autophagy [137,138]. As a result, they are highly susceptible to Cd-induced apoptosis. A study in rats has shown that Cd induced acute kidney injury through inhibiting autophagy and affecting lysosomal function [139]. Many other mechanisms have been proposed to explain how Cd causes tubular cell injury and death, discussed below.

#### 4.3.1. Mitochondrial Targeting

Cd reaches the inner membrane of mitochondria through MT and transport proteins for calcium and iron, including the metal coupling unit (MCU) and DMT1 [140]. There, Cd reduces the synthesis of adenosine triphosphate (ATP), suppresses the electron transport chain, and promotes the formation of reactive oxygen species (ROS), leading to mitochondrial injury, and mitochondrial DNA (mtDNA) is released [7,140,141,142]. This activates the nuclear factor-kappaB (NF-кB) and the DNA-sensing mechanism, cyclic GMP-AMP synthase–stimulator of interferon genes (cGAS-STING), signaling pathways. As a result, proinflammatory cytokines are released, and cell death ensues.

#### 4.3.2. Endoplasmic Reticulum Targeting

Cd-induced disruption of calcium homeostasis and tubular cell death has been demonstrated [143,144,145]. In primary rat proximal tubular cells, calcium release from the endoplasmic reticulum by Cd raised intracellular calcium concentrations and inhibited autophagy [143,144]. In mouse renal tubular cells, Cd affected the release and reuptake of calcium by the ER through the activation of calcium channels, namely, the phospholipase C (PLC)-inositol 1, 4, 5-trisphosphate receptor (IP3R) and sarco/endoplasmic reticulum Ca2^+^-ATPase (SERCA) [144]. In addition, the toxic Cd in mouse tubular epithelial cells was intensified via suppressing SERCA expression and decreasing the stability of the SERCA protein [145].

#### 4.3.3. Nuclear Targeting

As Figure 4 shows, Cd activates the transcription of MT and zinc transporter 1 (ZnT1) genes through its interaction with the metal response element-binding transcription factor-1 (MTF-1) [146,147]. In addition, Cd activates the transcription of the HO-1 gene through its interactions with a Cd response element (*CdRE*) and the Maf recognition antioxidant response element (*MARE*), also known as a stress response element (*StRE*), which are localized to the HO-1 promoter region [148]. Furthermore, Cd suppresses the lysosomal degradation of Nrf2 [149] and causes the nuclear export of the HO-1 gene repressor Bach1, which allows for the transactivation of the HO-1 gene by the Nrf2/small Maf complex [150].

Notably, the Cd-induced HO-1 expression is not coupled with bilirubin synthesis [151,152]. Takeda et al. (2015) used the eel fluorescent protein UnaG, which binds unconjugated bilirubin, to demonstrate, for the first time, that all cell types that they examined synthesized heme, from which bilirubin was continuously generated and released [151,152]. This de novo synthesis of heme is central to cellular homeostasis and the defense mechanism against oxidative stress damage. In addition, they found that Cd^2+^ and inorganic arsenic as As^3+^ increased HO-1 expression, but there was only a small change in bilirubin formation [151]. Thus, Cd induces HO-1 expression without a concomitant bilirubin synthesis. This knowledge is of significance to Cd toxicity research. It explains the pervasiveness of the toxicity of Cd as it can increase cellular oxidative stress, while at the same time, it shuts down the frontline cellular stress response, involving bilirubin synthesis. Indeed, Cd-induced HO-1 expression in renal tubular epithelial cells has now been shown to result in ferroptosis due to excessive mitochondrial ROS generation [153].

#### 4.3.4. Zinc Deficiency

As a result of an increased MT protein expression induced by Cd itself, most Cd in the cytosol is sequestered in MT, and there is only a small fraction of Cd localized at the basolateral membrane [154,155]. There is little evidence for an exit route for Cd once it has entered tubular cells. The metal efflux transporters ferroportin-1 (FPN1) and ZnT1 are highly specific for Fe and Zn, respectively [156,157,158,159].

The increased expression of both MT and ZnT1 by Cd produces fatal consequences, which include zinc loss; a limited availability of zinc for de novo heme biosynthesis; decreased heme catabolism; and the reduced catalytic activity of key antioxidant enzymes, namely, superoxide dismutase (SOD) and NADPH oxidase (Nox) [160,161]. ZnT1 is a unique efflux transporter that functions as a Zn/Ca exchanger [147,157,158,159], and as a mechanism to prevent toxicity from zinc overload, zinc is extruded from cells by ZnT1, and cellular zinc deficiency will follow.

Every nucleated cell in the body synthesizes heme, which is used as a substrate to generate bilirubin, a cytoprotective biomolecule [151]. Bilirubin is a potent antioxidant and a lipid peroxidation chain breaker [162]. HO-1 induction by zinc to ensure the synthesis of bilirubin is another universal antioxidant defense mechanism of zinc [158]. Using a reporter gene assay, zinc was shown to activate HO-1 gene expression via the antioxidant response element (ARE) and the nuclear factor (erythroid-derived 2)-like 2 (Nrf2) signaling pathway [163].

An enzyme in heme biosynthesis, named δ-aminolevulinic acid dehydratase (ALAD), requires zinc as a cofactor. Anemia due to zinc deficiency is caused by an insufficient amount of heme for hemoglobin synthesis [164]. Limited availability of zinc following MT and ZnT1 inductions by Cd causes a reduction in ALAD activity, which could eventually diminish the synthesis of bilirubin.

In summary, the ability of Cd to increase the expression of both ZnT1 and MT explains the development of zinc-deficiency conditions in Cd-intoxicated tubular cells. A similar phenomenon in CKD and its associated disease, CVD, is discussed below.

### 4.4. Zinc and Chronic Kidney Disease

A meta-analysis of data from 42 studies showed that CKD patients had lower serum zinc levels compared to controls [165]. In another meta-analysis of data from 15 randomized controlled trials, zinc supplementation was shown to reduce CVD risk in CKD patients [166,167]. Higher serum zinc concentrations were associated with lower risks of mortality from CVD in the Ludwigshafen Risk and Cardiovascular Health Study (Germany) [168].

In the NHANES 1988–1994 database, zinc intake below the RDA was associated with elevated urinary Cd excretion rates in both men and women [60]. Serum zinc levels of <74 μg/dL and blood Cd levels of >0.53 μg/L were associated with an increased risk of a low eGFR in the 2011–2012 NHANES cycle (n = 1545; aged ≥20 years) [61]. In a cohort of NHANES 2011–2018 participants, aged ≥ 18 years (n = 9557), equivalent to 236,263,413 community-dwelling U.S. adults, the top quartile of blood Cd was associated with a 2.79-fold increase in the risk of having CKD compared with the bottom quartile [169]. The risk of having CKD was reduced by 90% and 89%, comparing low zinc and high Cd with high zinc plus high Cd and high zinc plus low Cd, respectively.

The overall mean dietary Zn intake level among the NHANES 2003–2018 participants (n = 37,195) was 11.85 mg/day, and a dietary zinc intake of 16.46 mg/day was associated with a reduction in the prevalence of a low eGFR [170]. Of note, dietary Zn levels of 15–16 mg/day are higher than RDA values, and it is important to note that dietary zinc intake levels by participants displayed a U-shaped dose–response relationship [171]. Zinc intake levels lower than 6.64 mg/day or higher than 16 mg/day were associated with a higher CKD risk. Thus, zinc intake should not be too low (deficiency) or too high (overdose).

An experimental study observed rising systemic blood pressure and declining kidney function, measured by inulin clearance, in rats fed with a diet containing Zn that was 40 times higher than that of a normal diet for 4 weeks [168]. Thus, experimental data support the observation that a zinc overdose increased the risk of CKD in the representative U.S. population study [171].

### 4.5. Mitigation of Cadmium Nephrotoxicity in Chronic Kidney Disease

As the data in Table 1 indicate, Cd is found all food types, making exposure to Cd unavoidable. The current environmental Cd exposure has now reached toxic levels in a significant proportion of many populations. Thus, population exposure to environmental Cd presents global public health significance and many challenges. Alarmingly, CKD is predicted to become the fifth leading cause of years of life lost by 2040 should its rising prevalence continue [91]. Developing strategies to combat CKD and its progression to kidney failure are of global importance, given an immense associated healthcare cost.

#### 4.5.1. Population Zinc Supplementation

Zinc is an essential trace element, required for the function of 10% of proteins in the human body [172,173]. Because a storage mechanism for zinc does not exist, unlike iron, a sufficient daily supply is required to prevent deficiency [172]. According to the EFSA assessment, zinc requirement ranges from 7.5 to 12.7 mg/day for women and from 9.4 to 16.3 mg/day for men [174]. A dietary intake of zinc below the RDA is believed to be highly prevalent [175,176,177], but the health impact of marginal zinc intake is not easily recognized because no specific symptom can be attributable to zinc deficiency, and it cannot be readily measured [173,175,176,177].

The role of zinc in CKD development for any cause has been subjected to review [178,179,180]. Herein, findings from recent publications are summarized. Data from clinical trials indicate the efficacy of zinc supplementation in reducing the risk of CKD and mortality from CVD among patients with CKD [166,167,168]. Population zinc supplementation, however, has many challenges; zinc and iron deficiencies co-exist, especially in at-risk subpopulations (women of reductive age), and there are zinc–iron interactions, if supplemented together [24,25,172,180]. Furthermore, there is risk of anemia due to Cu deficiency induced by a high dose of zinc (≥80 mg/day), and an additional 2 mg of copper is required to prevent such anemia [181].

#### 4.5.2. Exogenous Heme Oxygenase-1 Inducers

The induction of HO-1 expression by phytochemicals can be considered as a potential mitigative strategy. In chronic environmental Cd exposure conditions, the metal binding sites of MT are saturated while other endogenous antioxidants, such as glutathione and bilirubin, are depleted. Like zinc, many plant biomolecules have been found to activate HO-1 expression through the stress response transcription factor Nrf2 and the Stress Responsive Element (StRE), known also as the Maf recognition antioxidant response element (MARE) [182].

As Figure 4 shows, an induced expression of HO-1 raises intracellular concentrations of bilirubin and carbon monoxide (CO). Bilirubin is a potent antioxidant, while CO inhibits cytochrome c oxidase (COX) activity and restores mitochondrial ROS signaling [182,183,184]. HO-1 induction thus provides tubular cells the capacity to resist Cd toxicity and ensures cell survival and normal functions are maintained. Kidney disease associated with chronic environmental Cd exposure is primarily due to oxidative damage to the tubule and zinc deficiency (Figure 4). This results in a sustained decline in eGFR, hypertension, and proteinuria (Section 3.2).

Green tea consumption increases HO-1 expression [185,186,187]. In a human trial that included diabetic subjects with no history of metabolic complications who did not smoke and did not take regular food supplements, the consumption of green tea in normal amounts increased HO-1 expression and reduced DNA damage in lymphocytes [187]. Other chemicals of plant origin known to increase HO-1 expression are curcumin, catechin (green tea), α-lipoic acid (broccoli and spinach), and sulforaphane (cruciferous vegetables) [188]. A study from Thailand observed a relationship between plant food scores and fasting blood sugar levels; higher plant food scores from the consumption of vegetables, fruits, nuts/seeds, and cereals were associated with lower fasting blood sugar levels [189]. The experimental studies below suggest that the anti-Cd toxicity of certain plant biomolecules could be attributable to HO-1 induction.

#### 4.5.3. Other Plant Biomolecules

Table 5 provides a summary of recently published preclinical studies of some plant biomolecules for alleviating the nephrotoxicity and/or hepatotoxicity of Cd.

## 5. Conclusions

Cd is present in nearly all food types, and as such, dietary exposure is inevitable for most people. Those with low body iron stores absorb Cd at a rate as high as 45%. From the gut, Cd reaches systemic circulation through specialized transport proteins, systems, and pathways for metal nutrients like zinc and iron. The kidney proximal tubule is the principal site of Cd accumulation and toxicity because it is well equipped with metal transporters and protein internalization mechanisms that facilitate Cd uptake.

Zinc deficiency has emerged as the cytotoxic mechanism of Cd. At very low concentrations, Cd induces the expression of an efflux transporter, ZnT1, that functions as a Zn/Ca exchanger, and consequently, zinc is extruded from cells by ZnT1. Zinc loss continues as Cd exposure persists. There is no equivalent exit route for Cd, and most acquired Cd is thus retained within tubular cells. Excreted Cd reflects cytotoxicity at the present time because it is released from injured or dying tubular cells.

Current evidence implicates chronic exposure to low-dose Cd does increase the worldwide prevalence of NAFLD, hypertension, and CKD. Incident CKD could be viewed as the signs of toxic environment continuance. CKD and hypertension in people chronically exposed to Cd arise primarily from proximal tubule damage, inflammation, tubulointerstitial fibrosis, tubular atrophy, and irreversible nephron destruction.

As inferred by NHANES data, a significant proportion of the general U.S. population will develop CKD from exposure to Cd in a normal diet, but the existing exposure limits for Cd, ranging between 0.21 and 0.83 µg/kg body weight per day, do not afford health protection. New health-protective exposure guidelines should be established.

By the benchmark dose response concept, a 5–10% decrease in eGFR can be used as a sensitive endpoint for estimating a safe dietary Cd exposure level, instead of β_2_M excretion exceeding 300 µg/g of creatinine. A 5–10% increase in total protein excretion could also be an early warning sign of Cd nephrotoxicity that is suitable for estimating a dietary Cd exposure level that produces a discernable impact on the kidneys.

Of concern, a significant proportion of many populations has a toxic Cd body burden. A permissible dietary Cd exposure level should preserve the functional integrity of the liver and kidneys while minimizing the risk of developing hypertension. Public health measures should also be developed to help minimize Cd contamination of food chains. The loss of eGFR in Cd-exposed people is irreversible. Furthermore, effective chelation therapy to remove Cd from the body does not exist, and treatment options when CKD reaches its end stage are limited. Thus, it appears pivotal to avoid foods containing high levels of Cd and to stop smoking. It is pivotal to maintain an optimal body content of metal nutrients, especially zinc and iron, which reduce Cd absorption rates, block its entrance into systemic circulation, and prevent zinc deficiency induced by the metal. Ensuring an adequate intake of dietary antioxidants is a complementary preventive measure. 

## Figures and Tables

**Figure 1 biomolecules-15-00183-f001:**
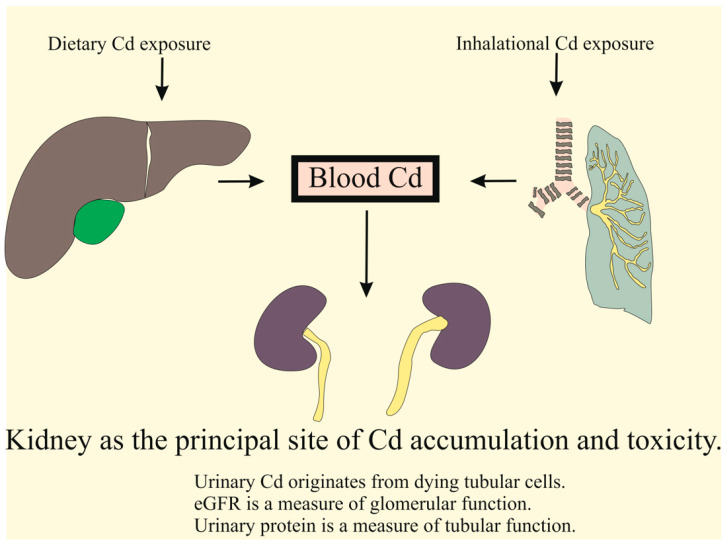
Exposure sources of cadmium and its pathway to kidney accumulation. From the gut, Cd ions are delivered to the liver via the portal system. The Cd ions that are not taken up by hepatocytes in the first pass reach systemic circulation and are delivered to cells of tissues and organs throughout the body. Cd can enter most cells in the body through the transport proteins for metal nutrients, namely, iron (Fe), calcium (Cd), zinc (Zn), copper (Co), manganese (Mn), and cobalt (Co).

**Figure 2 biomolecules-15-00183-f002:**
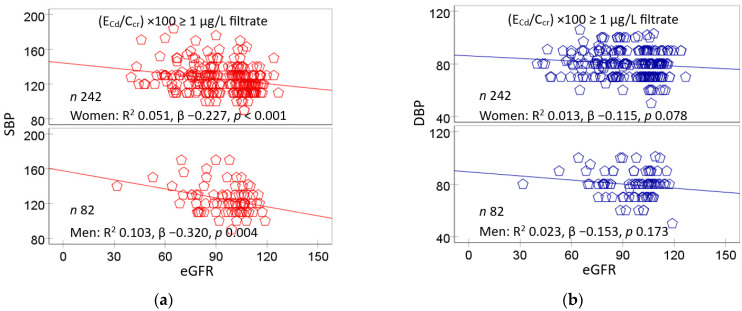
Inverse relationships of blood pressure and eGFR. Scatterplots relate SBP (**a**) and DBP (**b**) to eGFR in women and men. A high Cd burden was defined as an E_Cd_/C_cr_ value of ≥0.01 µg/L of filtrate.

**Figure 3 biomolecules-15-00183-f003:**
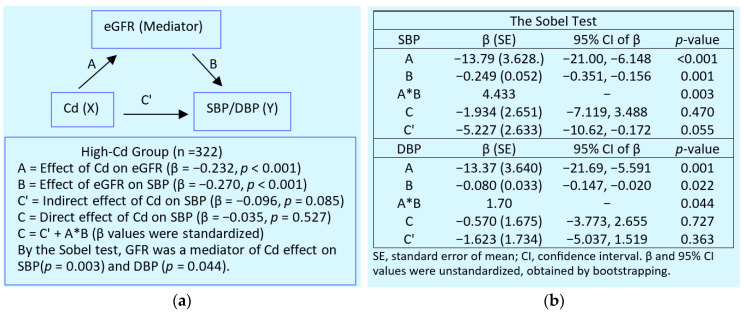
An analysis of mediating effects of cadmium. (**a**) A model that depicts eGFR as a mediator of Cd effects on SBP/DBP. (**b**) The Sobel test for the indirect effects of Cd on SBP/DBP.

**Figure 4 biomolecules-15-00183-f004:**
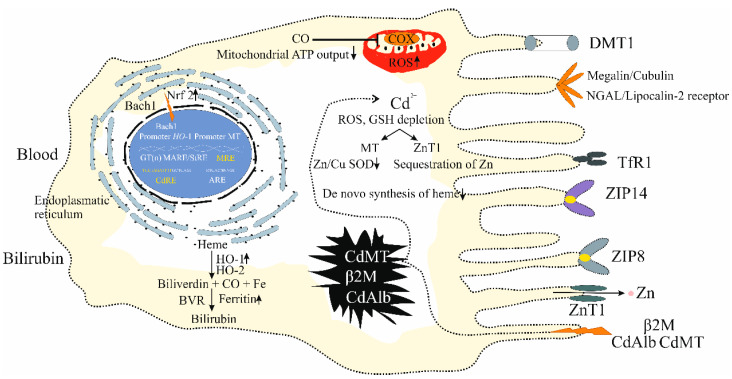
The kidney proximal tubular epithelial cell as the principal target of cadmium. The entry of Cd into tubular cells is through DMT1, ZIP8, ZIP10, ZIP14, the megalin/cubilin, and NGAL/lipocalin 2-mediated endocytosis. Abbreviations: DMT1, divalent metal transporter 1; ZIP8, Zrt- and Irt-related protein 8; ZIP10, Zrt- and Irt-related protein 10; ZIP14, Zrt- and Irt-related protein 14; NGAL, neutrophil gelatinase-associated lipocalin.

**Table 1 biomolecules-15-00183-t001:** Permissible content of cadmium in various foodstuffs.

Item	Specific Food Types	MPC ^a^ mg Cd/kg Wet Weight
1.	Meat (excluding offal) of bovine animals, sheep, pigs, and poultry.	0.05
2.	Horsemeat, excluding offal.	0.20
3.	Liver of bovine animals, sheep, pigs, poultry, and horses.	0.05
4.	Kidney of bovine animals, sheep, pigs, poultry, and horses.	1.0
5.	Muscle meat of fish, excluding species listed in #6 and #7.	0.05
6.	Bonito (Sarda sarda), common two-banded seabream (*Diplodus vulgaris*), eel (*Anguilla anguilla*), grey mullet (*Mugil labrosus labrosus*), horse mackerel or scad (*Trachurus* spp), louvar or luvar (*Luvarus imperialis*), mackerel (*Scomber* spp.), sardine (*Sardina pilchardus*), sardinops (*Sardinops* spp), tuna (*Thunnus* spp., *Euthynnus* spp., *Katsuwonus pelamis*), and wedge sole (*Dicologoglossa cuneata*).	0.10
7.	Muscle meat of bullet tuna (*Auxis* spp.).	0.20
8.	Muscle meat of anchovy (*Engraulis* spp) and swordfish (*Xiphias gladius*).	0.30
9.	Crustaceans, excluding the brown meat of crab and excluding the head and thorax meat of lobster and similar large crustaceans (*Nephropidae* and *Palinuridae*).	0.50
10.	Bivalve mollusks.	1.0
11.	Cephalopods (without viscera).	1.0
12.	Cereals, excluding bran, germ, wheat, and rice.	0.10
13.	Bran, germ, wheat, and rice.	0.20
14.	Soybeans.	0.20
15.	Vegetables and fruit, excluding leaf vegetables, fresh herbs, fungi, stem vegetables, root vegetables, and potatoes.	0.05
16.	Stem vegetables, root vegetables, and potatoes, excluding celeriac. For potatoes, the maximum level applies to peeled potatoes.	0.10
17.	Leaf vegetables, fresh herbs, celeriac, and the following fungi: *Agaricus bisporus* (common mushroom), *Pleurotus ostreatus* (Oyster mushroom), and *Lentinula edodes* (Shiitake mushroom).	0.20
18.	Fungi, excluding those listed in #17.	1.0
19.	Food supplements, excluding The food supplements listed in #20.	1.0
20.	Food supplements consisting exclusively or mainly of dried seaweed or of products derived from seaweed.	3.0

^a^ Maximum permissible level of cadmium in foods according to the European Food Safety Authority [18].

**Table 2 biomolecules-15-00183-t002:** Metal transport proteins involved in the absorption of cadmium from the gut to accumulate in kidney tubular cells.

Metal Transport Proteins	Cell Type/Localization	Physiological/Toxicological Roles
SLC39A14 (ZIP14)	Enterocyte/basolateral membrane	Transports Fe into and exits from enterocytes [23,24,25]. Transports Zn to tight junctions, especially in the jejunum for maintenance of the intestinal barrier function [29,30]. May mediate Cd absorption [31].
SLC11A2 (DMT1)	Enterocyte/apical membrane	Transports Fe into enterocytes and has the same high affinity for Cd as it has for Fe (Km 0.5~1 μM), with a high abundance in the duodenum [32,33]. Contributes to Cd absorption [34,35].
ATP7A	Enterocyte/trans–Golgi network, cytosol, and basolateral membrane	Transports Cu into portal blood, and ATP7A mutations are associated with Menkes disease [21]. May contribute to Cd absorption [36].
TRPV5 and TRPV6	Enterocyte/apical membrane	Transport Ca into enterocytes [37,38] and may provide Cd an entry route into enterocytes [39,40].
Calbindin-D9k	Enterocyte cytoplasm	Transports Ca to the basolateral membrane and its extrusion into portal blood [38,41]. Expression in ileum is induced by 1,25-dihydroxycholecalciferol [42]. May contribute to Cd absorption [43].
The NGAL/lipocalin 2 receptor system	Enterocyte/apical membrane	Assimilation of proteins and CdMT and CdPC complexes [26,27,28].
SLC39A8 (ZIP8)	Tubular epithelium/apical membrane	Uptake of Zn, Mn, and Cd [18,19].
The megalin/cubilin receptor system	Proximal tubule epithelium/apical membrane	Internalization of proteins, notably, albumin, β_2_M, and transferrin [44,45,46,47,48].
The NGAL/lipocalin 2 receptor system	The distal tubule and collecting duct epithelium	Internalization of proteins which may include CdMT [49,50].

SLC, solute-linked carrier; ZIP14, Zrt- and Irt-related protein 14; ZIP8, Zrt- and Irt-related protein 8; DMT1, divalent metal transporter 1; ATP7A, copper-transporting ATPases (Cu-ATPases); NGAL, neutrophil gelatinase-associated lipocalin; TRPV5, transient receptor potential vanilloid 5; TRPV6, transient receptor potential vanilloid 6.

**Table 3 biomolecules-15-00183-t003:** Environmental cadmium exposure levels associated with increased mortality and disease of the liver and kidneys.

Study Design/Population	Observed Effects and Cadmium Exposure Levels	Reference
Prospective; NHANES, 2005–2018; n = 8017; aged ≥20 years. Mortality data collection as of December 31, 2019.	^a^ HR (95% CI) values for all-cause mortality were 1.11 (0.85, 1.46); 1.42 (1.1, 1.84); and 1.67 (1.30, 2.13), comparing a urinary Cd of 0.116−0.231, 0.232−0.455, and > 0.455 µg/L with a urinary Cd of <0.116 µg/L.	Zhang et al., 2024 [83]
Prospective; NHANES, 1999–2014. A cohort of 1825 adults with CKD. Follow-up period, 6.8 years.	HR (95% CI) values for all-cause mortality were 1.75 (1.28, 2.39) and 1.59 (1.17, 2.15) at urinary Cd levels of ≥0.60 μg/g of creatinine and blood Cd levels of ≥0.70 μg/L, respectively.	Zhang et al., 2023 [15]
Prospective; NHANES, 2003−2012. A cohort of 24,810 adults; mean age, 44.4. Median follow-up period, 11.8 years.	Respective HR (95% CI) values for all-cause mortality among CKD and non-CKD cases were 1.42 (1.07, 1.88) and 1.40 (1.24, 1.58) at blood Cd levels of ≥0.4 μg/L.	Kuo et al., 2024 [84]
Prospective; U.S. adult participants of the Multi-Ethnic Study of Atherosclerosis; n = 6599; 53% female; mean age, 62.1 years. Followed from 2000–2001 through December 2019.	Respective HR (95% CI) values for incident CVD and all-cause mortality were 1.25 (1.03, 1.53) and 1.68 (1.43, 1.96), comparing Cd excretion rates of >0.80 with <0.35 µg/g of creatinine. Linear dose–response relationships were observed for both outcomes.	Martinez-Morata et al., 2024 [85]
Cross-sectional; NHANES, 1999–2020; n = 55,677; 20−85 years; 5175 (9.3%) had CKD.	OR values for CKD rose 2.1-fold, 3.2-fold, and 5.5-fold as blood Cd rose from <0.21 to 0.21–0.35, 0.36–0.60, and >0.60 µg/L, respectively. The reported increase in the risk of CKD due to Cd exposure was adjusted for smoking effects.	Akinleye et al., 2024 [12].
Cross-sectional; NHANES, 1988 –1994; n = 12,732; aged ≥20 years.	OR for liver inflammation rose 1.26-fold in women with a urinary Cd excretion of ≥ 0.83 μg/g of creatinine. Respective OR values for liver inflammation, NAFLD, and NASH rose 2.21-fold, 1.30-fold, and 1.95-fold in men with Cd excretion rates of ≥0.65 μg/g of creatinine.	Hyder et al., 2013 [86]
Cross-sectional; NHANES, 1999–2018; n = 47,422; aged ≥20 years.	OR for ^b^ advanced liver fibrosis rose 1.81-fold among those with blood Cd in the top quartile. This risk was found across racial/ethnic groups; Hispanic Blacks, Mexican Americans, and non-Hispanic Whites.	Ma et al., 2023 [87]

NHANES, National Health and Nutrition Examination Survey; HR, hazard ratio; CI, confidence interval; OR, odds ratio; NAFLD, non-alcoholic fatty liver disease; NASH, non-alcoholic steatohepatitis. ^a^ HR values were adjusted for age, sex, ethnicity, marital status, education, poverty–income ratio, depression, drinking, and NHANES cycles. ^b^ Advanced liver fibrosis was defined as a Fibrosis-4 of ≥2.67 and/or a Forns index of ≥6.9 and an abnormally high plasma level of alanine aminotransferase [86].

**Table 4 biomolecules-15-00183-t004:** A fall in eGFR and a concomitantly rising blood pressure are early effects of cadmium exposure.

Pathology	Consequence	Observation
Tubular cell injury.	Mild to moderate tubular dysfunction. Repair and regeneration.	A slight fall in eGFR and a slight elevation in blood pressure. Increased excretion of KIM-1.
Tubulointerstitial inflammation.	Nephron obstruction with cellular debris. Repair and regeneration.	A further fall in eGFR and a sustained increase in blood pressure. Increased excretion of NAG, β_2_M, and RBP.
Tubulointerstitial fibrosis and tubular atrophy.	Destruction of post-glomerular peritubular capillaries. Amputation of glomeruli from tubules.	Hypertension, proteinuria, and albuminuria and a further fall in eGFR.
Net loss of tubular cells per nephron. Glomerular atrophy.	CKD onset. Severe tubular dysfunction and tubular proteinuria.	When β_2_M excretion exceeds 300 µg/g of creatinine, eGFR will fall at a high rate. A rapid progression to end-stage kidney disease will ensue.

eGFR, estimated glomerular filtration rate; KIM-1, kidney injury molecule-1; NAG, N-acetyl-β-D-glucosaminidase; β_2_M, β_2_-microglobulin; RBP, retinol-binding protein.

**Table 5 biomolecules-15-00183-t005:** Preclinical investigations on anti-cadmium effects of some phytochemicals.

Therapeutic Target/Test Entity	Test Results	Reference
Kidney/Hyperin Quercetin-3-O-galactoside (flavonol glycoside)	Reduced Cd accumulation and attenuated Cd effects on mitochondria, apoptosis, and inflammation. Activated the Nrf-2/Keap-1 ARE pathway.	Lucky et al., 2024 [190]
Kidney/Linalool Monoterpene essential oils	Reduced histopathological lesions, inflammation, oxidative stress, and apoptosis.	Kaya and Yalçın, 2024 [191]
Kidney/Icariin 8-isopentenyl flavonoid glycoside	Reduced oxidative stress damage to tubular cells and their regeneration. Inhibited inflammasome formation. Activated the Toll-like receptor 4/P2rx7/nuclear factor kappa B (TLR4/P2rx7/NF-κB/NLRP3) signaling pathway.	Zheng et al., 2024 [192]
Kidney/Pinostrobin Flavonoid from Boesenbergia rotunda	Reduced the mitochondrial membrane potential and ameliorated Cd effects on TCA cycle enzymes and mitochondrial electron transport chain enzymes, such as succinate dehydrogenase, NADH dehydrogenase, cytochrome c-oxidase, and coenzyme Q-cytochrome reductase.	Ijaz et al., 2023 [193]
Kidney/*Physalis peruviana* L. calyx extract	Decreased TNF-α and NF-κβ levels. The molecular docking data suggest withanolides ^a^ may inhibit κB kinase activity.	Soliman et al., 2023 [194]
Kidney and liver/ chocolate	Reduced DNA damage and apoptotic and necrotic cell death and restored the mitochondrial membrane potential and the mitochondrial DNA copy number. Increased HO-1 and iNOS expression.	Mohamed, 2022 [195]
Kidney/Helianthemum lippii extract nanoparticles	Reduced kidney fibrosis, inflammatory cell infiltration, glomerular destruction, and tubular dilatation.	Laib et al., 2024 [196]
Kidney/Diallyl disulfide	Suppressed NF-κB, CD68, and pro-inflammatory mediators; attenuated oxidative stress and inflammation; suppressed TGF-β1/Smad3 signaling; and enhanced Nrf2/HO-1 signaling, antioxidants, and PPARγ.	Alruhaimi et al., 2024 [197]
Liver/Diallyl disulfide	Attenuated oxidative stress and apoptosis, suppressed TLR-4/NF-κB signaling, suppressed inflammation and oxidative stress, and upregulated PPARγ.	Alruhaimi et al., 2024 [198]
Liver/Morin 3,5,7,29,49-pentahydroxyflavone	Reduced ER stress; increased SOD, GSH, Gpx, CAT, Nrf2, IL-10, and IL-4; reduced TNF-α, IL-1-β, and IL-6; retarded apoptotic cascades; and suppressed JNK and p-PERK. Modulated upstream p-GRP78/PERK/ATF6 pro-apoptotic oxidative/ER stress and downstream JNK/BAX/caspase apoptotic signaling pathways.	Sengul et al., 2024 [199]

HO-1, heme oxygenase-1; iNOS, inducible nitric oxide synthase; ER, endoplasmic reticulum; SOD, superoxide dismutase; GSH, glutathione; MDA, malondialdehyde; PPARγ, peroxisome proliferator-activated receptor gamma; TNF-α, tumor necrosis factor-alpha; NF-κβ, nuclear factor kappaB. ^a^ β-hydroxywithanolide, physalin B, and 3α,14β dihydroxywithaphysalin.

## Data Availability

Not applicable.

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
