# Peer review of "Antioxidative Function of Zinc and Its Protection Against the Onset and Progression of Kidney Disease Due to Cadmium"

_biomolecules, 2025, doi:10.3390/biom15020183_

Round 1
Reviewer 1 Report
Comments and Suggestions for Authors
The manuscript reviews literature data showing that cadmium uptake is responsible for the onset of chronic kidney disease (CDK). The role of cadmium accumulation in leading to other metals dyshomeostasis, such as zinc, copper, and iron is also extensively reviewed. Most important, the author emphasizes the widespread presence of cadmium in cigarette smoke and in most kinds of food, as well as molecular pathway of cadmium intracellular uptake, leading to CDK.
In my opinion, the topic is of the outmost importance, since cadmium contamination is ofter underestimated, while it represents a major health threat, causing cancer and neurodegenerative diseases, as well as kidney failure. I suggest that the author include a table, showing cadmium contamination levels in different food; I feel this might help understanding that cadmium is easily uptaken by anybody.
I highly recommend accepting the paper.
minor points:
line 315 - "As Figure 4 presents" substitute with "As Figure 4 shows"
line 411 - Substitute "smoker" with "smoke"
Author Response
Reviewer 1.
Comments and Suggestions
The manuscript reviews literature data showing that cadmium uptake is responsible for the onset of chronic kidney disease (CDK). The role of cadmium accumulation in leading to other metals dyshomeostasis, such as zinc, copper, and iron is also extensively reviewed. Most important, the author emphasizes the widespread presence of cadmium in cigarette smoke and in most kinds of food, as well as molecular pathway of cadmium intracellular uptake, leading to CDK.
In my opinion, the topic is of the outmost importance, since cadmium contamination is ofter underestimated, while it represents a major health threat, causing cancer and neurodegenerative diseases, as well as kidney failure.
I suggest that the author include a table, showing cadmium contamination levels in different food; I feel this might help understanding that cadmium is easily uptaken by anybody.
I highly recommend accepting the paper.
Response: I thank the reviewer for the suggestion to improve a manuscript. Accordingly, a new Table 1 has been inserted to show permissible levels of Cd in various food types (lines 80-83). Changes to the text are in blue.
Minor points:
Line 315 - "As Figure 4 presents" substitute with "As Figure 4 shows"
Line 411 - Substitute "smoker" with "smoke"
Response: The referred typo errors have been corrected.

Reviewer 2 Report
Comments and Suggestions for Authors
The work by Mr. Satarug presents a comprehensive overview of Zn influence on Cd induced renal toxicity. The paper is sufficiently thorough but some sections should be explained in more detail. I attached the manuscript with detailed comments.

Author Response
Reviewer 2
Comments and Suggestions
The work by Mr. Satarug presents a comprehensive overview of Zn influence on Cd induced renal toxicity. The paper is sufficiently thorough but some sections should be explained in more detail. I attached the manuscript with detailed comments.
Response: Thank you for the comments and suggestions to improve a manuscript. All issues the reviewer raised have now been addressed. Changes to the text are in blue.
